# Multiobjective Distribution Matching

Xiaoyuan Zhang[1]  Peijie Li[2]  Yingying Yu[1]  Yichi Zhang[3]  Han Zhao[4]  Qingfu Zhang[1]

## Abstract

Distribution matching is a core concept in machine learning, with applications in generative models, domain adaptation, and algorithmic fairness. A closely related but less explored challenge is generating a distribution that aligns with multiple underlying distributions, often with conflicting objectives, known as a Pareto optimal distribution. In this paper, we develop a general theory based on information geometry to construct the Pareto set and front for the entire exponential family under KL and inverse KL divergences. This formulation allows explicit derivation of the Pareto set and front for multivariate normal distributions, enabling applications like multiobjective variational autoencoders (MOVAEs) to generate interpolated image distributions. Experimental results on real-world images demonstrate that both algorithms can generate high-quality interpolated images across multiple distributions.

## 1. Introduction

Distribution Matching (DM) is a fundamental concept in machine learning and has rich applications across multiple applications, including generative modeling (Goodfellow et al., 2014; Ho and Ermon, 2016; Li et al., 2015), domain adaptation (Baktashmotlagh et al., 2016; Ganin et al., 2016; Gong et al., 2024; Tachet des Combes et al., 2020; Zhao et al., 2018), causal representation learning (Johansson et al., 2016; Shalit et al., 2017), and algorithmic fairness (Zhang et al., 2018; Zhao et al., 2019b; 2022), just to name a few. A typical DM problem is formulated as:

$$\min_{\boldsymbol{\theta}} D(p_{\boldsymbol{\theta}} \| p),$$

where $p$ denotes the target distribution, and $p_{\boldsymbol{\theta}}$ is the distribution which needs to be optimized. Alternatively, the problem

[1]Department of Computer Science, CityUHK. [2]Department of Mathematics, HKU. [3]Department of Statistics, IU. [4]Department of Computer Science, UIUC. Correspondence to: Qingfu Zhang <qingfu.zhang@cityu.edu.hk>.

*Proceedings of the 42$^{nd}$ International Conference on Machine Learning*, Vancouver, Canada. PMLR 267, 2025. Copyright 2025 by the author(s).

can also be expressed in its inverse form: $\min_{\boldsymbol{\theta}} D(p \| p_{\boldsymbol{\theta}})$, since the divergence $D(\cdot \| \cdot)$ is not necessarily symmetric. This classical distribution matching problem has been extensively studied, with various approaches proposed for calculating the divergence such as the Wasserstein distance $\mathcal{W}(p_{\boldsymbol{\theta}} \| p)$ (Villani et al., 2009) and $f$-divergence Minimization (Nowozin et al., 2016). A widely used but underexplored problem addressed in this paper is aligning a single distribution with multiple distributions simultaneously, formulated as:

$$\begin{aligned}
\min_{\boldsymbol{\theta}} \boldsymbol{f}(\boldsymbol{\theta}) &= (f_1(\boldsymbol{\theta}), \ldots, f_m(\boldsymbol{\theta})) \\
&= (D(p_{\boldsymbol{\theta}} \| p_1), \ldots, D(p_{\boldsymbol{\theta}} \| p_m)),
\end{aligned} \tag{1}$$

where each objective can also be presented in its inverted form, $D(p_i \| p_{\boldsymbol{\theta}})$. MODM has broad applications in machine learning, including the controllable generation of intermediate distributions (e.g., images, drugs, speeches) between multiple underlying distributions, the development of models that balance multiple domain adaptations (Han and Pimentel, 2024; Jin et al., 2020; Wu et al., 2021), multi-source domain adaptation (Wen et al., 2020; Zhao et al., 2018), and group fairness with multiple sub-populations (Chen et al., 2023; Xian and Zhao, 2024; Xian et al., 2023).

This paper presents two approaches to study the MODM problem. The first approach assumes that the distributions $p_\theta, p_1, \ldots, p_m$ belong to a specific distribution family, such as the Exponential family. Under this condition, we first investigate a more general case: how to generate Pareto solutions when the decision space is a dually flat manifold endowed with a Riemannian metric. Building on these general results, we characterize the Pareto set for multivariate normal (MVN) distributions under both Kullback-Leibler (KL) divergence and inverse KL divergence as illustrative examples. Furthermore, we highlight a direct application of generating Pareto-optimal MVN distributions in the context of multiobjective Variational Autoencoder (MOVAE) algorithm. This algorithm learns multiple latent MVN distributions and employs non-linear decoders to map them into complex real-world applications. Experimental results demonstrate the effectiveness of the proposed method in this context. The contributions of this paper are summarized as follows:

1. We investigate the geometric structure of the multiobjective distribution matching problem with the tools of information geometry. We derive the explicit form of Pareto set on the dually flat manifold under the canonical divergence. Then, we give the shape of PS for the entire exponential family and use multivariate normal (MVN) distributions as a specific example.

2. Besides exponential family, we derive the form of the Pareto set under the $\alpha$-divergence $D^\alpha(\cdot\|\cdot)$. Based on our theoretical results, we design an algorithm called multiobjective variational auto-encoder (MOO-VAE) to generate interpolated images between multiple distributions.

3. We evaluate the performance of the proposed multiobjective distribution matching method not only on synthetic distributions but also on real-world image distributions. The proposed method can generate high-quality image distributions among multiple underlying distributions.

## 2. Related Work

Since this paper is both related with gradient-based multiobjective optimization (MOO) and distribution matching. We briefly discuss those two related topics separately.

### 2.1. Gradient-based MOO

Gradient-based MOO methods gained growing popularity with the successful application of Multiple Gradient Descent Algorithm (MGDA) (Sener and Koltun, 2018) in deep multitask learning. Methods to find the Pareto set (PS) using two strategies: identifying a diverse set of Pareto solutions or modeling the entire PS directly. Notable works include Pareto Multi-Task Learning (PMTL) (Lin et al., 2019), finding solutions restricted in specific objective regions; Exact Pareto Optimization (EPO) (Mahapatra and Rajan, 2020; 2021), Weighted Chebyshev (WC)-MGDA (Momma et al., 2022), and Preference-based MGDA (PMGDA) (Zhang et al., 2024), which align objectives with preference vectors; and Gradient-based Hypervolume maximization (HV-Grad (Deist et al., 2020; 2021; Emmerich et al., 2007)), which optimizes the hypervolume of a set of solutions to achieve both Pareto optimality and diversity. The most common approach, however, remains optimizing some aggregation functions that convert a multiobjective optimization problem (MOP) into a single-objective one. Besides finding a single Pareto optimal solution or a set of Pareto solutions, another popular gradient-based MOO paradigm is Pareto set learning (Chen and Kwok, 2024; Dimitriadis et al., 2023; Navon et al., 2020; Ruchte and Grabocka, 2021), which trains a single model to predict the entire PS, typically using a hypernetwork or a neural network with a low-rank adaptation structure.

### 2.2. Distribution matching

Distribution matching (DM), or distribution alignment, is a core concept in machine learning with broad applications, including domain adaptation (Ganin et al., 2016; Nguyen et al., 2023; Xiao et al., 2024; Zhang et al., 2022; Zhao et al., 2018; 2019a), algorithmic fairness (Prost et al., 2019; Quadrianto and Sharmanska, 2017; Zhao and Gordon, 2022; Zhao et al., 2019b), and generative models (e.g., GANs, VAEs, diffusion models, imitation learning) (Higgins et al., 2017; Ho and Ermon, 2016; Jin et al., 2020; Yin et al., 2023).

For generative models like GANs, the objective is to generate a distribution that closely aligns with the data distribution. Similarly, in generative adversarial imitation learning (GAIL) (Ho and Ermon, 2016), the goal is to learn a policy such that the state-action density function matches that of the expert. DM also help learn a representation that aligns two distributions and has been applied to enhance robustness and enforce constraints in domain generalization, causal discovery, and fair representation learning (Gong et al., 2024).

## 3. Preliminaries

### 3.1. Multiobjective optimization

For a MOP (Equation (1)), it is difficult to compare the quality of solutions since vector objectives do not admit a total order. To describe the optimality for a MOP, we first introduce the concept of Pareto optimality.

**Definition 1** (Pareto Optimal (PO), Pareto Set (PS), Pareto Front (PF)). *A solution $\boldsymbol{\theta}^*$ is PO if no other solution $\boldsymbol{\theta}' \in \Theta$ dominates it, denoted as $\boldsymbol{f}(\boldsymbol{\theta}') \preceq_{\text{strict}} \boldsymbol{f}(\boldsymbol{\theta}^*)$, i.e., $f_i(\boldsymbol{\theta}') \leq f_i(\boldsymbol{\theta}^*)$ for all $i \in [m]$ with at least one strict inequality hold. The set of all PO solutions is called the Pareto set, and its image is the Pareto front.*

In addition to PO solutions, *weakly* PO solutions are those that cannot be *strictly* dominated by the other solutions, i.e., no solution $\boldsymbol{\theta}'$ exists such that $\forall i \in [m]$, $f_i(\boldsymbol{\theta}') < f_i(\boldsymbol{\theta}^*)$, or $\boldsymbol{f}(\boldsymbol{\theta}') \prec \boldsymbol{f}(\boldsymbol{\theta}^*)$. The simplest way to find PO solutions is to use aggregation functions $g_{\boldsymbol{\lambda}}(\cdot) : \mathbb{R}^m \mapsto \mathbb{R}$ to convert a vector optimization problem into a scalar one. Widely-used aggregation functions include linear scalarization, $g_{\boldsymbol{\lambda}}(\boldsymbol{f}(\boldsymbol{\theta})) = \sum_{i=1}^m \lambda_i f_i(\boldsymbol{\theta})$ and Tchebycheff functions, $g_{\boldsymbol{\lambda}}(\boldsymbol{f}(\boldsymbol{\theta})) = \max_{i \in [m]} \lambda_i(f_i(\boldsymbol{\theta}) - z_i)$, where $\boldsymbol{z}$ is a reference point such that $\boldsymbol{z} \preceq \boldsymbol{f}(\boldsymbol{\theta})$, for $\boldsymbol{\theta} \in \Theta$. When applying aggregation functions to solve a MOP, we have the following lemma for the optimal solution of an aggregation function.

**Lemma 2** (Adapted from (Miettinen, 1999), Theorem 2.6.2). *If $g_{\boldsymbol{\lambda}}(\boldsymbol{f}(\boldsymbol{\theta}))$ is decreasing w.r.t, $\boldsymbol{f}(\boldsymbol{\theta}))$, i.e., $\boldsymbol{f}(\boldsymbol{\theta}^{(a)}) \preceq_{\text{strict}} \boldsymbol{f}(\boldsymbol{\theta}^{(b)})$, $g_{\boldsymbol{\lambda}}(\boldsymbol{f}(\boldsymbol{\theta}^{(a)})) \leq g_{\boldsymbol{\lambda}}(\boldsymbol{f}(\boldsymbol{\theta}^{(b)}))$, the optimal solution of $g_{\boldsymbol{\lambda}}(\boldsymbol{f}(\boldsymbol{\theta}))$ is a weakly PO solution. If $g_{\boldsymbol{\lambda}}(\boldsymbol{f}(\boldsymbol{\theta}))$*

*is strictly decreasing w.r.t, $\boldsymbol{f}(\boldsymbol{\theta})$, i.e., when $\boldsymbol{f}(\boldsymbol{\theta}^{(a)}) \preceq_{\text{strict}} \boldsymbol{f}(\boldsymbol{\theta}^{(b)})$, $g_{\boldsymbol{\lambda}}(\boldsymbol{f}(\boldsymbol{\theta}^{(a)})) < g_{\boldsymbol{\lambda}}(\boldsymbol{f}(\boldsymbol{\theta}^{(b)}))$, the optimal solution of $g_{\boldsymbol{\lambda}}(\boldsymbol{f}(\boldsymbol{\theta}))$ is a PO solution.*

Lemma 2 guarantees that, optimizing aggregation functions with a specific preference vector can find a (weakly) Pareto optimal solution. The remaining issue is whether the entire PS can be recovered by using all preference vectors that span the probability simplex. For this question we know that (1) when each objective function $f_i(\boldsymbol{\theta})$ is convex, for any Pareto solution, there exists a preference vector where the optimal solution is the result of the linear aggregation function under this preference vector (Boyd, 2004)[Section 4.7], (2) for any weakly PO solution, there exists a preference vector such that the optimal value of the Tchebycheff scalarization function corresponds to this solution (Choo and Atkins, 1983). Under mild conditions (c.f. (Zhang et al., 2023)[Prop. 2]), optimizing the Tchebycheff function yields the "exact" Pareto solution, where the optimal solution $\boldsymbol{\theta}^*$ of the Tchebycheff function satisfies: $\lambda_1(f_1(\boldsymbol{\theta}^*) - z_1) = \ldots = \lambda_m(f_m(\boldsymbol{\theta}^*) - z_m)$.

Another popular MOO paradigm is called *Pareto Set Learning (PSL)* (Lin et al., 2020; 2022), which learns a model $t_\beta(\boldsymbol{\lambda}) : \Delta_m \mapsto \text{PS}$ that maps a preference vector to a PO solution. Different from previous methods, PSL aims to learn the entire PS with a single model rather than to find a finite set of Pareto solutions.

### 3.2. Information geometry

This section introduces some basic concepts of information geometry (Amari, 2016; Ay et al., 2017; Nielsen, 2020), begin with the concept of Riemannian manifold.

**Definition 3** (Riemannian manifold (Lee, 2012)). *A Riemannian manifold is a pair $(S, g)$, where $S$ is a smooth manifold locally resembling the Euclidean space, and $g : p \mapsto \langle \cdot, \cdot \rangle_p$ is a Riemannian metric that equips each tangent space $T_p(S)$ with an inner product $\langle \cdot, \cdot \rangle_p : T_p(S) \times T_p(S) \mapsto \mathbb{R}_+$.*

A connection on a Riemannian manifold describes how to differentiate vector fields.

**Definition 4** (Connection of a smooth manifold (Lee, 2012)). *A connection $\nabla$ on a manifold $S$ is a bilinear map on the space of smooth vector fields $\Gamma(S)$:*

$$\nabla : \Gamma(S) \times \Gamma(S) \mapsto \Gamma(S),$$

*satisfying the following properties:*

*1. $\nabla_{fX} Z = f \nabla_X Z$,*

*2. $\nabla_X (fY) = f \nabla_X Y + (Xf)Y$,*

*for vector fields $X, Y, Z \in \Gamma(S)$ and smooth function $f \in C^\infty(S)$, where $Xf$ is the derivative of $f$ along $X$.*

Given a (local) coordinate system $\boldsymbol{\vartheta} = [\vartheta^i]$ on $S$, denoted by $\{\partial_i := \frac{\partial}{\partial \vartheta^i}\}$ the coordinate vector fields, the metric $g$ can be characterized by the smooth functions $g_{ij} = \langle \partial_i, \partial_j \rangle$ (the metric components), and the connection $\nabla$ can be characterized by the smooth functions $\Gamma_{ij,k} = \langle \nabla_{\partial_i} \partial_j, \partial_k \rangle$ (the Christoffel symbols). A connection $\nabla$ is said to be *flat* if there exists a coordinate system $\boldsymbol{\vartheta}$ such that the corresponding Christoffel symbols of $\nabla$ are all zero, and $\boldsymbol{\vartheta}$ is said to be an *affine coordinate system* of the flat connection $\nabla$.

Statistical models $S = \{p_{\boldsymbol{\vartheta}}\}_{\boldsymbol{\vartheta} \in \Theta}$, that is, parametrized probability distributions (density functions), are considered smooth manifolds in information geometry. Moreover, these statistical manifolds are often endowed with the *Fisher information metric* $g^F$ given by

$$g_{ij}^F(\boldsymbol{\vartheta}) := \mathbb{E}_{\boldsymbol{\vartheta}} \left[ \partial_i \ell_{\boldsymbol{\vartheta}} \cdot \partial_j \ell_{\boldsymbol{\vartheta}} \right],$$

and the $\alpha$-*connections* $\nabla^{(\alpha)}$ given by

$$\Gamma_{ij,k}^{(\alpha)}(\boldsymbol{\vartheta}) := \mathbb{E}_{\boldsymbol{\vartheta}} \left[ \left( \partial_i \partial_j \ell_{\boldsymbol{\vartheta}} + \frac{1-\alpha}{2} \partial_i \ell_{\boldsymbol{\vartheta}} \cdot \partial_j \ell_{\boldsymbol{\vartheta}} \right) \partial_k \ell_{\boldsymbol{\vartheta}} \right],$$

where $\ell_{\boldsymbol{\vartheta}} := \log p_{\boldsymbol{\vartheta}}$ and $\mathbb{E}_{\boldsymbol{\vartheta}}[f] := \int f(\boldsymbol{x}) p_{\boldsymbol{\vartheta}}(\boldsymbol{x}) \, \mathrm{d}\boldsymbol{x}$. Next, we introduce dualistic structures on statistical manifolds, essential for analyzing information geometry.

**Definition 5** (Dual connection $\nabla^*$). *Let $(S, g)$ be a Riemannian manifold and $\nabla, \nabla^*$ be two connections on $S$. If*

$$Z \langle X, Y \rangle = \langle \nabla_Z X, Y \rangle + \langle X, \nabla_Z^* Y \rangle$$

*holds for all $X, Y, Z \in \Gamma(S)$, then we say that $\nabla$ and $\nabla^*$ are duals of each other with respect to $g$.*

Under a coordinate system, the condition for dual connection can be rewritten as

$$\partial_i g_{jk} = \Gamma_{ij,k} + \Gamma_{ik,j}^*.$$

Clearly, $\nabla^{(\alpha)}$ and $\nabla^{(-\alpha)}$ are dual with respect to $g^F$. In particular, the pair $\nabla^{(1)}$ and $\nabla^{(-1)}$ are of special interest. We call them *exponential connection* $\nabla^{(e)} := \nabla^{(1)}$ and *mixture connection* $\nabla^{(m)} := \nabla^{(-1)}$ respectively.

In this paper, we focus on dually flat manifolds $(S, g, \nabla, \nabla^*)$ where the pair of dual connections $\nabla$ and $\nabla^*$ on $(S, g)$ are both flat. On a dually flat manifold, we can always find a pair of dual coordinate systems $\boldsymbol{\vartheta} = [\vartheta^i], \boldsymbol{\eta} = [\eta_j]$, such that $\boldsymbol{\vartheta}$ and $\boldsymbol{\eta}$ are the affine coordinate systems of $\nabla$ and $\nabla^*$ respectively, and $\langle \partial_i, \partial^j \rangle = \delta_i^j$ for $\partial_i := \frac{\partial}{\partial \vartheta^i}, \partial^j := \frac{\partial}{\partial \eta_j}$. Moreover, we can also find a pair of potential functions $\psi, \varphi$ corresponding to $\boldsymbol{\vartheta}, \boldsymbol{\eta}$ such that $\partial_i \psi = \eta_i, \partial^i \varphi = \vartheta^i$ and $\psi + \varphi = \sum_i \vartheta^i \eta_i$. Under these settings, the *canonical divergence* of $(S, g, \nabla, \nabla^*)$, which is the key notion in our discussion, is then defined as

$$D(p \| q) := \psi(p) + \varphi(q) - \sum_i \vartheta^i(p) \eta_i(q).$$

Note that the canonical divergence is independent of the choice of dual coordinate systems and potential functions (Amari and Nagaoka, 2000). Dually flat manifolds are closely related to the widely-used Bregman divergences in machine learning.

**Definition 6** (Bregman divergence). *For a strictly convex smooth function $f$ defined on some open $\Xi \subseteq \mathbb{R}^n$, the corresponding Bregman divergence $D_f$ is defined as*

$$D_f(\boldsymbol{x}\|\boldsymbol{y}) := f(\boldsymbol{x}) - f(\boldsymbol{y}) - \sum_i \partial_i f(\boldsymbol{y})\left(x^i - y^i\right),$$

*for $\boldsymbol{x}, \boldsymbol{y} \in \Xi$, where $\partial_i f$ is the $i$-th partial derivative of $f$.*

Given a pair of dual coordinate systems $\boldsymbol{\vartheta}, \boldsymbol{\eta}$ and the corresponding potential functions $\psi, \varphi$ on a dually flat manifold $(S, g, \nabla, \nabla^*)$, the canonical divergence can be expressed as a Bregman divergence:

$$D(p\|q) = D_\psi(\boldsymbol{\vartheta}(p)\|\boldsymbol{\vartheta}(q)) = D_\varphi(\boldsymbol{\eta}(q)\|\boldsymbol{\eta}(p)).$$

Conversely, given a strictly convex smooth function $f$ on $\Xi \subseteq \mathbb{R}^n$, the Riemannian metric characterized by

$$g_{ij}(\boldsymbol{x}) := \partial_i \partial_j f(\boldsymbol{x}),$$

and the pair of affine connections characterized by

$$\Gamma_{ij,k}(\boldsymbol{x}) = 0, \quad \Gamma^*_{ij,k}(\boldsymbol{x}) = \partial_i \partial_j \partial_k f(\boldsymbol{x}),$$

form a dually flat structure on $\Xi$, with canonical divergence given by the Bregman divergence $D_f$. In the following, we present some examples of dually flat statistical manifolds.

**Example 7** (Exponential family). *An $n$-dimensional model $S = \{p_{\boldsymbol{\vartheta}}\}_{\boldsymbol{\vartheta} \in \Theta}$ is called an exponential family, if it can be expressed in terms of functions $\{C, F_1, \ldots, F_n\}$ on the base space and a function $\psi$ on $\Theta$ as*

$$p_{\boldsymbol{\vartheta}}(\boldsymbol{x}) = \exp\left[C(\boldsymbol{x}) + \sum_{i=1}^n \vartheta^i F_i(\boldsymbol{x}) - \psi(\boldsymbol{\vartheta})\right]$$

*with*

$$\psi(\boldsymbol{\vartheta}) = \log \int \exp\left[C(\boldsymbol{x}) + \sum_{i=1}^n \vartheta^i F_i(\boldsymbol{x})\right] \mathrm{d}\boldsymbol{x}.$$

*For an exponential family $S$, $(S, g^F, \nabla^{(e)}, \nabla^{(m)})$ is indeed a dually flat manifold. The $\nabla^{(e)}$-affine natural parameters $\boldsymbol{\vartheta} = [\vartheta^i]$ and the $\nabla^{(m)}$-affine expectation parameters $\boldsymbol{\eta} = [\eta_j]$ defined by $\eta_j(\boldsymbol{\vartheta}) := \mathbb{E}_{\boldsymbol{\vartheta}}[F_j]$ give a pair of dual coordinate systems. The corresponding potential functions are given by the cumulant function $\psi$ and*

$$\varphi(\boldsymbol{\vartheta}) = \sum_{i=1}^n \theta^i \eta_i(\boldsymbol{\vartheta}) - \psi(\boldsymbol{\vartheta}) = -H(p_{\boldsymbol{\vartheta}}) - \mathbb{E}_{\boldsymbol{\vartheta}}[C],$$

*where $H$ is the* Shannon entropy *given by*

$$H(p) := -\int p(\boldsymbol{x}) \cdot \log p(\boldsymbol{x}) \, \mathrm{d}\boldsymbol{x}.$$

*The canonical divergence is then given by*

$$\begin{aligned} D(p\|q) &= \sum_{i=1}^n \left(\vartheta^i(q) - \vartheta^i(p)\right) \eta_i(q) + \psi(p) - \psi(q) \\ &= \int \left(\log q(\boldsymbol{x}) - \log p(\boldsymbol{x})\right) q(\boldsymbol{x}) \, \mathrm{d}\boldsymbol{x} = D_{\mathrm{KL}}(q\|p), \end{aligned}$$

*which is indeed the dual of the* Kullback–Leibler divergence. *Many important probabilistic models belong to exponential families, such as multivariate normal distributions, chi-squared distributions, gamma distributions, Poisson distributions, multinomial distribution (with fixed trial), distributions on a finite space (probability simplex) ... In fact, arbitrarily given density functions $p_0, p_1, \ldots, p_n$, then*

$$p_{\boldsymbol{\vartheta}}(\boldsymbol{x}) := \frac{p_0(\boldsymbol{x})^{1-\sum_i \vartheta^i} p_1(\boldsymbol{x})^{\vartheta^1} \cdots p_n(\boldsymbol{x})^{\vartheta^n}}{\int p_0(\boldsymbol{x})^{1-\sum_i \vartheta^i} p_1(\boldsymbol{x})^{\vartheta^1} \cdots p_n(\boldsymbol{x})^{\vartheta^n} \, \mathrm{d}\boldsymbol{x}}$$

*gives an exponential family. Furthermore, (Banerjee et al., 2005) showed that any dually flat manifold can be realized as an exponential family (see also (Amari, 2016)).*

*Here we present the model of multivariate normal distributions (MVNs), which is widely used in machine learning:*

$$\begin{aligned} p_{\boldsymbol{\vartheta}}(\boldsymbol{x}) &= |2\pi\boldsymbol{\Sigma}|^{-\frac{1}{2}} \exp\left\{-\frac{1}{2}(\boldsymbol{x}-\boldsymbol{\mu})^\top \boldsymbol{\Sigma}^{-1}(\boldsymbol{x}-\boldsymbol{\mu})\right\} \\ &= \exp\left\{(\boldsymbol{\vartheta}^A)^\top \boldsymbol{F}_A(\boldsymbol{x}) + \mathrm{Tr}(\boldsymbol{\vartheta}^B F_B(\boldsymbol{x})) - \psi(\boldsymbol{\vartheta})\right\} \end{aligned}$$

*is expressed as an exponential family in terms of*

$$C(\boldsymbol{x}) = 0, \quad \boldsymbol{F}_A(\boldsymbol{x}) = \boldsymbol{x}, \quad \boldsymbol{F}_B(\boldsymbol{x}) = \boldsymbol{x}\boldsymbol{x}^\top,$$

*and*

$$\begin{aligned} \psi(\boldsymbol{\vartheta}) &= \frac{1}{2}\boldsymbol{\mu}^\top \boldsymbol{\Sigma}^{-1} \boldsymbol{\mu} + \frac{1}{2}\log|2\pi\boldsymbol{\Sigma}| \\ &= -\frac{1}{4}(\boldsymbol{\vartheta}^A)^\top(\boldsymbol{\vartheta}^B)^{-1}\boldsymbol{\vartheta}^A + \frac{1}{2}\log\left|-\pi(\boldsymbol{\vartheta}^B)^{-1}\right|, \end{aligned}$$

*with respect to the natural parameters $\boldsymbol{\vartheta} = [\boldsymbol{\vartheta}^A, \boldsymbol{\vartheta}^B]$:*

$$\boldsymbol{\vartheta}^A = \boldsymbol{\Sigma}^{-1}\boldsymbol{\mu}, \quad \boldsymbol{\vartheta}^B = -\frac{1}{2}\boldsymbol{\Sigma}^{-1},$$

*and the expectation parameters $\boldsymbol{\eta} = [\boldsymbol{\eta}^A, \boldsymbol{\eta}^B]$:*

$$\boldsymbol{\eta}_A = \boldsymbol{\mu}, \quad \boldsymbol{\eta}_B = \boldsymbol{\Sigma} + \boldsymbol{\mu}\boldsymbol{\mu}^\top.$$

**Example 8** (Mixture family). *An $n$-dimensional model $S = \{p_{\boldsymbol{\vartheta}}\}_{\boldsymbol{\vartheta} \in \Theta}$ is called a mixture family, if it can be expressed in terms of functions $\{C, F_1, \ldots, F_n\}$ on the base space as*

$$p_{\boldsymbol{\vartheta}}(\boldsymbol{x}) = C(\boldsymbol{x}) + \sum_{i=1}^n \vartheta^i F_i(\boldsymbol{x}).$$

*For a mixture family $S$, $(S, g^F, \nabla^{(m)}, \nabla^{(e)})$ is indeed a dually flat manifold. The $\nabla^{(m)}$-affine parameters $\boldsymbol{\vartheta} = [\vartheta^i]$ and the $\nabla^{(e)}$-affine parameters $\boldsymbol{\eta} = [\eta_j]$ defined by*

$$\eta_j(\boldsymbol{\vartheta}) := \int F_j(\boldsymbol{x}) \cdot \log p_{\boldsymbol{\vartheta}}(\boldsymbol{x}) \, d\boldsymbol{x}$$

*give a pair of dual coordinate systems. The corresponding potential functions are given by*

$$\psi(\boldsymbol{\vartheta}) = -H(p_{\boldsymbol{\vartheta}}), \quad \varphi(\boldsymbol{\vartheta}) = -\int C(\boldsymbol{x}) \cdot \log p_{\boldsymbol{\vartheta}}(\boldsymbol{x}) \, d\boldsymbol{x}.$$

*The canonical divergence is then given by*

$$D(p\|q) = \sum_{i=1}^{n} \vartheta^i(p) \left( \eta_i(p) - \eta_i(q) \right) + \varphi(q) - \varphi(p)$$

$$= \int p(\boldsymbol{x}) \left( \log p(\boldsymbol{x}) - \log q(\boldsymbol{x}) \right) d\boldsymbol{x} = D_{\mathrm{KL}}(p\|q),$$

*which is exactly the* Kullback–Leibler *divergence.*

*A frequently used formulation is the mixture of distributions: arbitrarily given density functions $p_0, p_1, \ldots, p_n$, then*

$$p_{\boldsymbol{\vartheta}}(\boldsymbol{x}) := \left( 1 - \sum_{i=1}^{n} \vartheta^i \right) p_0(\boldsymbol{x}) + \sum_{i=1}^{n} \vartheta^i p_i(\boldsymbol{x})$$

$$= p_0(\boldsymbol{x}) + \sum_{i=1}^{n} \vartheta^i \left( p_i(\boldsymbol{x}) - p_0(\boldsymbol{x}) \right)$$

*is expressed as a mixture family in terms of*

$$C(\boldsymbol{x}) = p_0(\boldsymbol{x}), \quad F_i(\boldsymbol{x}) = p_i(\boldsymbol{x}) - p_0(\boldsymbol{x}), \ i \in [n].$$

# 4. Multiobjective distribution matching theories

In this section, we first introduce the underlying geometry of multiobjective distribution matching (MODM) based on dually flat spaces, and later derive the explicit forms of PS and PF of MODM under some specific divergences.

## 4.1. Geometric structure of MODM

**Theorem 9.** *Let $(S, g, \nabla, \nabla^*)$ be a dually flat manifold, $\boldsymbol{\vartheta}$ and $\boldsymbol{\eta}$ be a pair of dual coordinate systems, and $D(p\|q)$ be the canonical divergence.*

- *The Pareto set of the MOP with the*

$$\min_{p \in S} \left( D(p_1\|p), D(p_2\|p), \ldots, D(p_m\|p) \right)$$

*is given by the convex hull of $p_1, \ldots, p_m$ enclosed by $\nabla$-geodesics, that is*

$$\left\{ p \in S : \boldsymbol{\vartheta}(p) = \sum_{k=1}^{m} \lambda_k \boldsymbol{\vartheta}(p_k), \boldsymbol{\lambda} \in \Delta_m \right\},$$

*where $\boldsymbol{\vartheta}(p)$ denote the coordinate of $p$ under the affine coordinate system $\boldsymbol{\vartheta}$ and the same for $\eta(p)$.*

- *The Pareto set of the MOP*

$$\min_{p \in S} \left( D(p\|p_1), D(p\|p_2), \ldots, D(p\|p_m) \right)$$

*is given by the convex hull of $p_1, \ldots, p_m$ enclosed by $\nabla^*$-geodesics, that is*

$$\left\{ p \in S : \boldsymbol{\eta}(p) = \sum_{k=1}^{m} \lambda_k \boldsymbol{\eta}(p_k), \boldsymbol{\lambda} \in \Delta_m \right\}.$$

The proof is postponed to Appendix A. The theorem builds that for non-degrade cases $(\dim(S) = n$ and $p_1, \ldots, p_k$ are linearly independent in corresponding coordinates), the PS is indeed an isomorphism of the $m$-dimensional simplex.

Next, we present the direct results for MVNs:

- For the MOP:

$$\min_{p \in \mathrm{MVNs}} \left( D_{\mathrm{KL}}(p\|p_1), D_{\mathrm{KL}}(p\|p_2), \ldots, D_{\mathrm{KL}}(p\|p_m) \right)$$

where $p_k$ is the MVN of $(\boldsymbol{\mu}_k, \boldsymbol{\Sigma}_k)$, the PS contains MVNs of $(\boldsymbol{\mu}, \boldsymbol{\Sigma})$ such that

$$\begin{cases} \boldsymbol{\Sigma}^{-1} \boldsymbol{\mu} = \displaystyle\sum_{k=1}^{m} \lambda_k \boldsymbol{\Sigma}_k^{-1} \boldsymbol{\mu}_k \\ \boldsymbol{\Sigma}^{-1} = \displaystyle\sum_{k=1}^{m} \lambda_k \boldsymbol{\Sigma}_k^{-1} \end{cases}, \ \boldsymbol{\lambda} \in \Delta_m. \qquad (2)$$

- For the MOP:

$$\min_{p \in \mathrm{MVNs}} \left( D_{\mathrm{KL}}(p_1\|p), D_{\mathrm{KL}}(p_2\|p), \ldots, D_{\mathrm{KL}}(p_m\|p) \right)$$

where $p_k$ is the MVN of $(\boldsymbol{\mu}_k, \boldsymbol{\Sigma}_k)$, the PS contains MVNs of $(\boldsymbol{\mu}, \boldsymbol{\Sigma})$ such that,

$$\begin{cases} \boldsymbol{\mu} = \displaystyle\sum_{k=1}^{m} \lambda_k \boldsymbol{\mu}_k \\ \boldsymbol{\Sigma} + \boldsymbol{\mu}\boldsymbol{\mu}^\top = \displaystyle\sum_{k=1}^{m} \lambda_k (\boldsymbol{\Sigma}_k + \boldsymbol{\mu}_k \boldsymbol{\mu}_k^\top) \end{cases}, \ \boldsymbol{\lambda} \in \Delta_m.$$

$$(3)$$

The corresponding PFs can then be obtained by computing the KL divergence between the MVNs on the PS and the given MVNs. Note that the KL divergence between the MVN $p_1$ of $(\boldsymbol{\mu}_1, \boldsymbol{\Sigma}_1)$ and the MVN $p_2$ of $(\boldsymbol{\mu}_2, \boldsymbol{\Sigma}_2)$ is given by

$$D_{\mathrm{KL}}(p_1\|p_2) = \frac{1}{2} \left[ D_\psi(\boldsymbol{\Sigma}_1^{-1}\|\boldsymbol{\Sigma}_2^{-1}) + D_{\boldsymbol{\Sigma}_2^{-1}}(\boldsymbol{\mu}_1, \boldsymbol{\mu}_2) \right],$$

where

$$D_\psi(\boldsymbol{\Sigma}_1^{-1}\|\boldsymbol{\Sigma}_2^{-1}) := \text{Tr}\left(\boldsymbol{\Sigma}_1\boldsymbol{\Sigma}_2^{-1}\right) - \log\left|\boldsymbol{\Sigma}_1\boldsymbol{\Sigma}_2^{-1}\right| - n$$

is a Bregman divergence between positive-definite matrices corresponding to $\psi(\boldsymbol{\Sigma}^{-1}) = -\log\left|\boldsymbol{\Sigma}^{-1}\right|$, and

$$D_{\boldsymbol{\Sigma}_2^{-1}}(\boldsymbol{\mu}_1, \boldsymbol{\mu}_2) := (\boldsymbol{\mu}_1 - \boldsymbol{\mu}_2)^\top \boldsymbol{\Sigma}_2^{-1}(\boldsymbol{\mu}_1 - \boldsymbol{\mu}_2)$$

is the (squared) Mahalanobis distance between $\boldsymbol{\mu}_1$ and $\boldsymbol{\mu}_2$ with respect to $\boldsymbol{\Sigma}_2^{-1}$. Note also that $\frac{1}{2}D_\psi(\boldsymbol{\Sigma}_1^{-1}\|\boldsymbol{\Sigma}_2^{-1})$ is exactly the KL divergence between the MVNs of covariant matrices $\boldsymbol{\Sigma}_1$ and $\boldsymbol{\Sigma}_2$ with the same mean, and $\frac{1}{2}D_{\boldsymbol{\Sigma}_2^{-1}}(\boldsymbol{\mu}_1, \boldsymbol{\mu}_2)$ is exactly the KL divergence between the MVNs of means $\boldsymbol{\mu}_1$ and $\boldsymbol{\mu}_2$ with the same covariant matrix $\boldsymbol{\Sigma}_2$.

Figures 1 and 2a illustrate the Pareto fronts (PFs) of MVN distributions with two and three objectives under KL divergence. In Figure 1, blue dots represent gradient descent solutions with $\boldsymbol{\Sigma}$ obtained via LU decomposition, while the red curve shows the theoretical PF from Theorem 9. The numerical results from gradient descent align well with the theoretical predictions. For completeness, we provide the PS for MVNs under Wassertein distance.

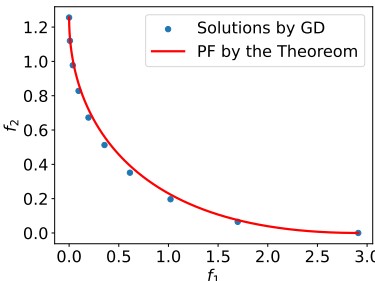

Figure 1. The PF of a 2-objective, 3-dimensional multivariate normal (MVN) distribution matching under the KL divergence, where $\boldsymbol{\mu}_1 = [0,0,0]^\top$, $\boldsymbol{\mu}_2 = [1,1,1]^\top$, $\boldsymbol{\Sigma}_1 = \text{diag}([1,1,1])$, and $\boldsymbol{\Sigma}_2 = \begin{bmatrix} 2.5 & -0.5 & 0 \\ -0.5 & 2.5 & 0 \\ 0 & 0 & 4 \end{bmatrix}$. Discrete solutions (in blue) are optimized by gradient descent (GD), and the theoretical PF (in red) curve is predicted by Theorem 9.

**Example 10** (PS under 2-Wasserstein distance $\mathcal{W}_2$). *The Wasserstein distance between two MVN distributions owns an explicit form:*

$$\mathcal{W}_2(p, p_k)^2 = \|\boldsymbol{\mu} - \boldsymbol{\mu}_k\|^2 + \text{Tr}\left[(\boldsymbol{\Sigma}^{1/2} - \boldsymbol{\Sigma}_k^{1/2})^2\right].$$

*The gradients w.r.t $\boldsymbol{\mu}$ and $\boldsymbol{\Sigma}$ are:*

$$\begin{cases} \dfrac{\partial\mathcal{W}_2(p, p_k)^2}{\partial\boldsymbol{\mu}} = 2(\boldsymbol{\mu} - \boldsymbol{\mu}_k), \\ \dfrac{\partial\mathcal{W}_2(p, p_k)^2}{\partial\boldsymbol{\Sigma}^{1/2}} = 2(\boldsymbol{\Sigma}^{1/2} - \boldsymbol{\Sigma}_k^{1/2}). \end{cases}$$

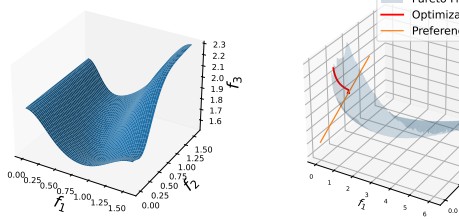

(a) PF of a 3-objective MODM problem. / (b) The optimization trajectory using Tchebycheff aggregation function.

Figure 2. Results on a 3-obj problem under KL divergence using parameters, $\boldsymbol{\mu}_1 = [0,0]^\top$, $\boldsymbol{\mu}_2 = [1,1]^\top$, $\boldsymbol{\mu}_3 = [2,2]^\top$, and $\boldsymbol{\Sigma}_1 = \begin{bmatrix} 1 & 0 \\ 0 & 1 \end{bmatrix}$, $\boldsymbol{\Sigma}_2 = \begin{bmatrix} 1 & 1 \\ 1 & 3 \end{bmatrix}$, $\boldsymbol{\Sigma}_3 = \begin{bmatrix} 5 & 2 \\ 2 & 4 \end{bmatrix}$.

*The PS can be formulated as:*

$$\begin{cases} \boldsymbol{\mu} = \displaystyle\sum_{k=1}^m \lambda_k\boldsymbol{\mu}_k \\ \boldsymbol{\Sigma}^{1/2} = \displaystyle\sum_{k=1}^m \lambda_k\boldsymbol{\Sigma}_k^{1/2} \end{cases}, \quad \boldsymbol{\lambda} \in \Delta_m. \qquad (4)$$

Equations (2) to (4) provide a principled approach for merging multiple distributions, which is also closely related to recently proposed model merging techniques (Yang et al., 2024; Zeng et al., 2025). Our results show that, under certain divergences, the entire Pareto set can be recovered through appropriate manipulations in the parametric space—specifically, by manipulating mean vectors and variance matrices.

### 4.2. MODM on the preference simplex

In this section, we provide a result of knowing the exact from of the PS of the MODM problem. Once we know the of the PS, it is easy to locate a specific Pareto solution under a preference vector. The following theorem shows that "optimizing over the entire probability space" is equivalent to optimizing on the preference simplex.

**Theorem 11.** *For a decreasing aggregation function, optimizing it on the PS is equivalent to optimizing it over the entire decision space $\Theta$.*

For example, with MVN distributions under KL divergence, optimizing a solution in $\Theta$ reduces to optimizing within a smaller $m$-dim space $\Delta_m$:

$$\min_{\boldsymbol{\lambda}\in\Delta_m} g_{\boldsymbol{\lambda}}\left(\boldsymbol{f}\left(\sum_{i=1}^m \lambda_i\boldsymbol{\theta}_i\right)\right) \iff \min_{\boldsymbol{\theta}\in\Theta} g_{\boldsymbol{\lambda}}(\boldsymbol{f}(\boldsymbol{\theta})).$$

*Proof.* Assuming $\boldsymbol{\theta}^* \in (\Theta - \mathrm{PS})$, but there exists a solution $\boldsymbol{\theta}' \in \mathrm{PS}$ such that $\boldsymbol{f}(\boldsymbol{\theta}') \preceq_{\mathrm{strict}} \boldsymbol{f}(\boldsymbol{\theta}^*)$, which contradicts a decreasing aggregation function. $\square$

These results show that MODM with aggregation functions reduces to a single-objective optimization with a simplex constraint, while the original problem involves more parameters and requires costly semi-definite programming due to the positive definite constraint on $\boldsymbol{\Sigma}$. Additionally, $g_{\boldsymbol{\lambda}}$ is convex w.r.t $\boldsymbol{\lambda}$, and the decision space, an $m$-dim simplex, is convex. The gradient of $g_{\boldsymbol{\lambda}}$ under KL divergence is:s

$$\frac{\partial}{\partial \lambda_k} g_{\boldsymbol{\lambda}} \left( \boldsymbol{f} \left( \sum_{i=1}^m \lambda_i \boldsymbol{\theta}_i \right) \right) = \nabla g_{\boldsymbol{\lambda}} \cdot \nabla \boldsymbol{f} \left( \sum_{i=1}^m \lambda_i \boldsymbol{\theta}_i \right) \cdot \boldsymbol{\theta}_k$$

which is convex with respect to $\boldsymbol{\lambda}$. The proof is in Appendix B. Optimizing the modified Tchebycheff function on the simplex using projective gradient descent (Algorithm 1) yields the "exact" Pareto solution, allowing precise control over its position. The optimization curve is shown in Figure 2b.

---

**Algorithm 1** MODM on the Preference Simplex.

1: **Initialization:** The initial preference vector $\boldsymbol{\lambda} \in \Delta_m$, where $\Delta_m$ is the preference simplex.
2: **for** epoch $= 1$ **to** $N_{\mathrm{epoch}}$ **do**
3:     $\boldsymbol{\lambda} \leftarrow \boldsymbol{\lambda} - \eta \cdot \nabla_{\boldsymbol{\lambda}} g_{\boldsymbol{\lambda}} \left( \boldsymbol{f} \left( \sum_{i=1}^m \lambda_i \boldsymbol{\theta}_i \right) \right)$
4:     $\boldsymbol{\lambda} \leftarrow \mathrm{Proj}_{\Delta_m}(\boldsymbol{\lambda})$
5: **end for**

---

### 4.3. The Pareto set under $\alpha$-divergence

We first recap that, The $\alpha$-divergence is defined as:

$$D^\alpha(p\|q) = \frac{4}{1-\alpha^2} \left( 1 - \int p(\boldsymbol{x})^{\frac{1-\alpha}{2}} q(\boldsymbol{x})^{\frac{1+\alpha}{2}} \, d\boldsymbol{x} \right).$$

It recovers the KL divergence at $\alpha \to -1$ and the reverse KL divergence at $\alpha = 1$, making it a special case of $f$-divergence.

**Theorem 12.** *For the MOP:*

$$\min_p \left( D^\alpha(p_1\|p), D^\alpha(p_2\|p) \ldots, D^\alpha(p_m\|p) \right)$$

*over all distributions, the PS is given by:*

$$p_{\boldsymbol{\lambda}} = \left[ \frac{1}{\psi(\boldsymbol{\lambda})} \left( \sum_{k=1}^m \lambda_k p_k^{\frac{1-\alpha}{2}} \right) \right]^{\frac{2}{1-\alpha}}, \quad \boldsymbol{\lambda} \in \Delta_m, \quad (5)$$

*where*

$$\psi(\boldsymbol{\lambda}) = \left[ \int \left( \sum_{k=1}^m \lambda_k p_k(\boldsymbol{x})^{\frac{1-\alpha}{2}} \right)^{\frac{2}{1-\alpha}} d\boldsymbol{x} \right]^{\frac{1-\alpha}{2}}.$$

*Here we use $p_{\boldsymbol{\lambda}}$, $p_1, \ldots, p_m$ as shorthand notations for the density functions of the distributions $p_{\boldsymbol{\lambda}}$, $p_1$, and $p_2$ on the underlying space, respectively.*

*Proof.* Fix $\boldsymbol{\lambda} \in \Delta_m$, consider the convex combination:

$$f(p) = \sum_{k=1}^m \lambda_k D^\alpha(p_k\|p) \tag{6}$$

Then we have the difference between $f(p)$ and $f(p_{\boldsymbol{\lambda}})$ can be formulated as:

$$f(p) - f(p_{\boldsymbol{\lambda}}) = \sum_{k=1}^m \lambda_k (D^\alpha(p_k\|p) - D^\alpha(p_k\|p_{\boldsymbol{\lambda}})) \tag{7}$$

$$= \frac{4}{1-\alpha^2} \int \left( \sum_{k=1}^m \lambda_k p_k^{\frac{1-\alpha}{2}} \right) \left( p_{\boldsymbol{\lambda}}^{\frac{1+\alpha}{2}} - p^{\frac{1+\alpha}{2}} \right) d\boldsymbol{x} \tag{8}$$

If we set $p_{\boldsymbol{\lambda}}$ to be the following form

$$p_{\boldsymbol{\lambda}}^{\frac{1-\alpha}{2}} = \frac{1}{\psi(\boldsymbol{\lambda})} \sum_{k=1}^m \lambda_k p_k^{\frac{1-\alpha}{2}}, \tag{9}$$

then by replacing the term $p_{\boldsymbol{\lambda}}^{\frac{1-\alpha}{2}}$ in Equation (8) we have,

$$\begin{aligned} &f(p) - f(p_{\boldsymbol{\lambda}}) \\ &= \frac{4}{1-\alpha^2} \cdot \psi(\boldsymbol{\lambda}) \int p_{\boldsymbol{\lambda}}^{\frac{1-\alpha}{2}} (p_{\boldsymbol{\lambda}}^{\frac{1+\alpha}{2}} - p^{\frac{1+\alpha}{2}}) \, d\boldsymbol{x} \\ &= \psi(\boldsymbol{\lambda}) \cdot \frac{4}{1-\alpha^2} \left( 1 - \int p_{\boldsymbol{\lambda}}^{\frac{1-\alpha}{2}} p^{\frac{1+\alpha}{2}} \, d\boldsymbol{x} \right) \\ &= \psi(\boldsymbol{\lambda}) \cdot D^\alpha(p_{\boldsymbol{\lambda}}\|p) \geq 0. \end{aligned}$$

Thus, $p_{\boldsymbol{\lambda}}$ minimizes $f(p)$, and the minimum is given by

$$\begin{aligned} f(p_{\boldsymbol{\lambda}}) &= \sum_{k=1}^m \lambda_k D^\alpha(p_k\|p_{\boldsymbol{\lambda}}) \\ &= \frac{4}{1-\alpha^2} \left( 1 - \int \left( \sum_{k=1}^m \lambda_k p_k^{\frac{1-\alpha}{2}} \right) p_{\boldsymbol{\lambda}}^{\frac{1+\alpha}{2}} \, d\boldsymbol{x} \right) \\ &= \frac{4}{1-\alpha^2} \left( 1 - \psi(\boldsymbol{\lambda}) \right) \end{aligned}$$

$\square$

**Example 13.** *Especially for $\alpha = \pm 1$, we have*

- *When $\alpha = -1$, $D^{-1} = D_{\mathrm{KL}}$. Hence for the MOP:*

$$\min_p \left( D_{\mathrm{KL}}(p_1\|p), D_{\mathrm{KL}}(p_2\|p) \ldots, D_{\mathrm{KL}}(p_m\|p) \right)$$

*over all distributions, the PS is given by:*

$$p_{\boldsymbol{\lambda}} = \sum_{k=1}^m \lambda_k p_k, \quad \boldsymbol{\lambda} \in \Delta_m.$$

*Furthermore,*

$$\sum_{k=1}^{m} \lambda_k D_{\mathrm{KL}}(p_k \| p_{\boldsymbol{\lambda}}) = D_{\mathrm{JS}}^{(\boldsymbol{\lambda})}(p_1 \| \cdots \| p_m),$$

*which is the $\boldsymbol{\lambda}$-skew Jensen-Shannon divergence.*

- *When $\alpha = 1$, $D^1 = D_{\mathrm{KL}}^*$. Hence for the MOP:*

$$\min_{p} \left( D_{\mathrm{KL}}(p\|p_1), D_{\mathrm{KL}}(p\|p_2) \ldots, D_{\mathrm{KL}}(p\|p_m) \right)$$

*over all distributions, the PS is given by:*

$$p_{\boldsymbol{\lambda}} = \exp\left[ D_{\mathrm{B}}^{(\boldsymbol{\lambda})}(p_1\| \cdots \|p_m) + \sum_{k=1}^{m} \lambda_k \ln p_k \right],$$

*where $D_{\mathrm{B}}^{(\boldsymbol{\lambda})}(p_1\| \cdots \|p_m) := -\ln \int p_1^{\lambda_1} \cdots p_m^{\lambda_m}$ is the $\boldsymbol{\lambda}$-skew Bhattacharyya divergence. Furthermore,*

$$\sum_{k=1}^{m} \lambda_k D_{\mathrm{KL}}(p_{\boldsymbol{\lambda}}\|p_k) = D_{\mathrm{B}}^{(\boldsymbol{\lambda})}(p_1\| \cdots \|p_m).$$

### 4.4. Multiobjective Varational AutoEncoders (MOVAEs)

The MOVAE algorithm has two parts, which is shown in Algorithm 2 in Appendix C. During training, a VAE model is trained with all samples from multiple distribution with shared encoder ($\phi_i$) and decoder ($\psi_i$) parameters. After passing through the encoder network, each image is converted into a multivariate normal distribution.

During inference, for any preference vector, a Pareto-optimal distribution is generated using Equation (2), Equation (3) or Equation (4). This interpolated MVN distribution is then input to the decoder network. MOVAE's core idea is that directly generating PO distributions is challenging, so encoders and decoders transform normal distributions into real-world image distributions. Generating PO distribution and then converting it to other distributions approximates the generation of PO distributions. Another advantage of MOVAE over VAEs is that it requires only a single model, as Pareto MVN distributions in the hidden layers are computed using explicit formulations, avoiding the need for neural models. In contrast, mixture models with a controlling ratio $\boldsymbol{\lambda}$ require training separate models for each $\boldsymbol{\lambda}$. Therefore, the proposed method is more efficient in training and storage compared to VAEs with mixture distributions.

## 5. Experiments

In this section, we present the results of MOVAE. Results for various tradeoff levels are shown in Figure 3 under the inverse KL divergence. For other results, please refer to Appendix D. The size of the image is 28. Both the encoder and

decoder networks have around 157K parameters. Number of training images is around 12K. The optimizer is Adam with a learning rate of 3e-5.

The inference results of Figure 3 demonstrate that MOVAE is able to generate smooth interpolations between the (alarm,clock) distribution and the other circle distribution. We use five uniform preferences from $[1, 0]$ to $[0, 1]$ as examples, though any preference within this range can serve as input to the MOVAE network. By taking the preference from $[1, 0]$ to $[0, 1]$, the interpolated images gradually resembles the second distribution. For an intermediate preference, MOO generate a blending distribution of images under this preference.

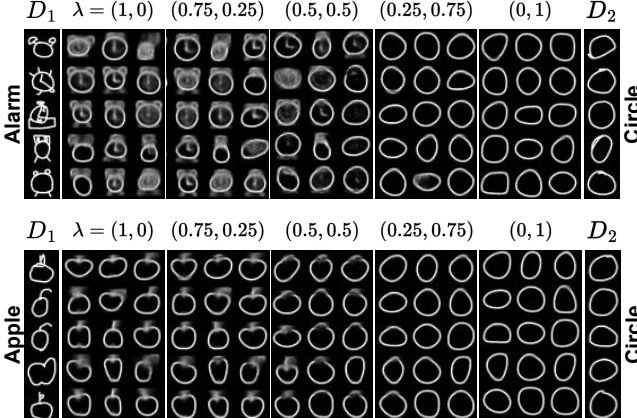

*Figure 3.* MOVAE Results under inverse KL divergence.

## 6. Conclusion, further work, and limitations

**Conclusion.** This paper explores the less-studied MODM by modeling a general multiobjective optimization problem on the dually flat Riemannian manifold. This approach provides explicit formulations for the entire exponential family with KL and inverse KL divergence. We also discuss the explicit form of the PS under $\alpha$-divergence. The theoretical results have direct applications, including multiobjective variational autoencoders (MOVAE). We evaluate MOVAE on the QuickDraw dataset, demonstrating their ability to generate blending images across different preference vectors.

**Future work.** Beyond the geometry explored here, distribution matching and information geometry share an intricate connection. Future work will address advanced topics, such as projecting distributions outside a manifold onto the manifold.

**Limitations.** This paper mainly focus on the theoretical results on multiobjective distribution matching and do not

discuss much on the applications. In the future, we will discuss the relationship between multiobjective distribution matching LLM, trustworthy machine learning and other topics.

## Acknowledgement

This work was supported by the Research Grants Council of Hong Kong, GRF Project No. CityU 11212524.

## Impact Statement

This paper presents work whose goal is to advance the field of multiobjective optimization and its applications in distribution matching. Given the scope of this research, we do not anticipate immediate ethical concerns or direct societal consequences. Therefore, we believe there are no specific ethical considerations or immediate societal impacts to be emphasized in the context of this work.

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

## A. Proof of Theorem 9

*Proof.* Due to the following property of the potential functions:

$$\frac{\partial \varphi}{\partial \boldsymbol{\eta}} = \boldsymbol{\vartheta}, \quad \frac{\partial \psi}{\partial \boldsymbol{\vartheta}} = \boldsymbol{\eta},$$

the gradients of $\frac{\partial \varphi}{\partial \boldsymbol{\eta}}$ or $\frac{\partial \varphi}{\partial \boldsymbol{\vartheta}}$ are given by:

$$\begin{aligned}
\frac{\partial}{\partial \boldsymbol{\eta}(p)} D(p_k \| p) &= \frac{\partial}{\partial \boldsymbol{\eta}(p)} \left( \psi(p_k) + \varphi(p) - \boldsymbol{\eta}(p)^\top \boldsymbol{\vartheta}(p_k) \right) \\
&= \frac{\partial \varphi}{\partial \boldsymbol{\eta}}(p) - \boldsymbol{\vartheta}(p_k) \\
&= \boldsymbol{\vartheta}(p) - \boldsymbol{\vartheta}(p_k)
\end{aligned}$$

$$\begin{aligned}
\frac{\partial}{\partial \boldsymbol{\vartheta}(p)} D(p \| p_k) &= \frac{\partial}{\partial \boldsymbol{\vartheta}(p)} \left( \psi(p) + \varphi(p_k) - \boldsymbol{\vartheta}(p)^\top \boldsymbol{\eta}(p_k) \right) \\
&= \frac{\partial \psi}{\partial \boldsymbol{\vartheta}}(p) - \boldsymbol{\eta}(p_k) \\
&= \boldsymbol{\eta}(p) - \boldsymbol{\eta}(p_k)
\end{aligned}$$

Hence for any $\boldsymbol{\lambda} \in \Delta_m$, we have

$$\sum_{k=1}^m \lambda_k \frac{\partial}{\partial \boldsymbol{\eta}(p)} D(p_k \| p) = 0 \iff \boldsymbol{\vartheta}(p) = \sum_{k=1}^m \lambda_k \boldsymbol{\vartheta}(p_k)$$

$$\sum_{k=1}^m \lambda_k \frac{\partial}{\partial \boldsymbol{\vartheta}(p)} D(p \| p_k) = 0 \iff \boldsymbol{\eta}(p) = \sum_{k=1}^m \lambda_k \boldsymbol{\eta}(p_k)$$

which completes the proof. □

## B. Proof of Theorem 11

If $f(\boldsymbol{x})$ is a convex function, then $f\left(\sum_{i=1}^m \lambda_i x_i\right)$ is also convex with respect to the weights $\lambda_i$, provided that $\boldsymbol{\lambda}$ belongs to the simplex $\Delta = \{\boldsymbol{\lambda} \in \mathbb{R}^m \mid \lambda_i \geq 0, \sum_{i=1}^m \lambda_i = 1\}$. Given that $f(x)$ is convex, for any points $x_1, \ldots, x_m$ and any set of weights $\lambda_1, \ldots, \lambda_m$, the following inequality holds:

$$f\left(\sum_{i=1}^m \lambda_i x_i\right) \leq \sum_{i=1}^m \lambda_i f(x_i).$$

To examine the convexity of $f\left(\sum_{i=1}^m \lambda_i x_i\right)$ with respect to $\boldsymbol{\lambda}$, let us consider $\boldsymbol{\lambda}^a$ and $\boldsymbol{\lambda}^b$ as two distinct points in the simplex, and $k_1, k_2 \geq 0$ such that $k_1 + k_2 = 1$. Then, we have:

$$f\left(k_1 \sum_{i=1}^m \lambda_i^a x_i + k_2 \sum_{i=1}^m \lambda_i^b x_i\right) \leq k_1 f\left(\sum_{i=1}^m \lambda_i^a x_i\right) + k_2 f\left(\sum_{i=1}^m \lambda_i^b x_i\right).$$

This inequality demonstrates that $f\left(\sum_{i=1}^m \lambda_i x_i\right)$ is convex with respect to the decision variables $\boldsymbol{\lambda}$. Essentially, the convexity of $f$ over its argument translates into the convexity of the function with respect to the weight parameters $\lambda_i$, as long as $\boldsymbol{\lambda}$ remains within the simplex constraints.

Further since for any $f_i$ are convex, take $g_{\boldsymbol{\lambda}}$ is a non-decreasing convex function, the overall function $g_{\boldsymbol{\lambda}}(\boldsymbol{f}(\sum_{i=1}^m \lambda_i x_i))$ is convex.

## C. Algorithms

---

**Algorithm 2** Multiobjective VAE (MOVAE)

---

1: # Step 1. Offline VAE Training.
2: **Input:** $m$ datasets $\mathcal{D}_1, \ldots, \mathcal{D}_m$.
3: **for** epoch = $1 : N_{\text{epoch}}$ **do**
4:     Sample a batch size of data, $\hat{\mathcal{D}}_1, \ldots, \hat{\mathcal{D}}_m$.
5:     **for** $i = 1 : m$ **do**
6:         $\ell_i = \ell_{\text{BCE}} + D_{\text{KL}}(p(z \parallel \theta), \mathcal{N}(0, I))$.
7:     **end for**
8:     $\ell = \frac{1}{m} \sum_{i=1}^{m} \ell_i$.
9:     **for** $i = 1 : m$ **do**
10:         Update: $\phi_i \leftarrow \phi_i - \eta \frac{\partial \ell}{\partial \phi_i}$, $\psi_i \leftarrow \psi_i - \eta \frac{\partial \ell}{\partial \psi_i}$, where $\eta$ is the learning rate.
11:     **end for**
12: **end for**
13: # Step 2. MOVAE Prediction.
14: **Input:** $\{\theta_i^{(1)}\}_{i=1}^{K}, \ldots, \{\theta_i^{(m)}\}_{i=1}^{K}$ to obtain $m$ parameters $(\mu_1, \Sigma_1), \ldots, (\mu_m, \Sigma_m)$ from the encoder network.
15: Given a preference vector $\lambda$, generate the PO distribution with parameters using Equation (2),Equation (3) or Equation (4).
16: Given a preference vector $\lambda$, generate the PO distribution with parameters using Equation (2), Equation (3) or Equation (4).
17: **Output:** Image distribution under the Pareto optimal distribution.

---

# D. Extra experimental results

In this section, we present extra results under KL and Wasserstein distance in Figures 4 and 5.

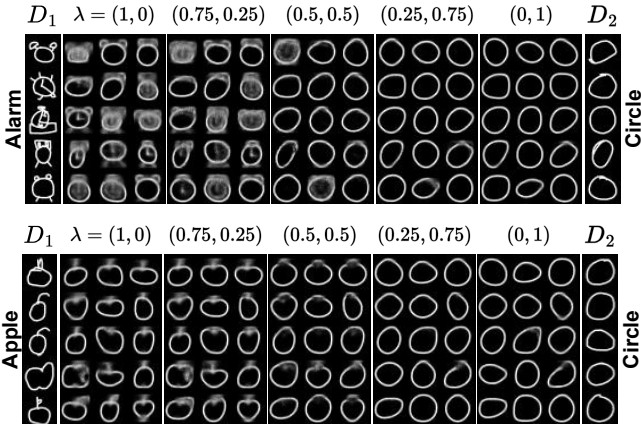

*Figure 4.* MOVAE Results under KL divergence.

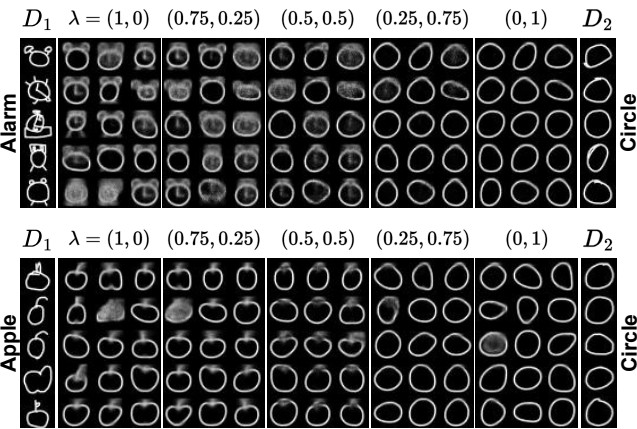

*Figure 5.* MOVAE Results under Wasserstein distances.

