# OpenReview forum: "Multiobjective distribution matching"
_ICML.cc/2025/Conference — ICML 2025 poster_

### Official Review · Reviewer_LE4u · 2025-03-13

**Overall Recommendation:** 2

**Summary:**

The paper tries to figure out how to do distribution matching to generate a distribution that aligns with multiple underlying distributions, often with conflicting objectives, known as a Pareto optimal distribution. The paper develops a theory on information geometry to construct the Pareto set. This allows for explicit derivation of the Pareto set and front for multivariate normal distributions. This leads to algorithms like multiobjective variational autoencoders (MOVAEs) to generate interpolated data distributions that can be used in multiple application fields. Based on the theory, the paper proposes the multiobjective generative adversarial network (MOGAN) algorithm, which is shown to be able to interpolate high quality real world images across domains.

**Claims And Evidence:**

**Claim #1**: "A related but less explored challenge is generating a distribution that aligns with multiple underlying distributions, often with conflicting objectives, known as a Pareto optimal distribution."
- This is supported by convincing evidence. The issue that using one distribution to align multiple distributions has really applications in machine learning.

**Claim #2**: "We figure out the difficulty of multiobjective distribution arises from the constrained parameter space and the complex geodesics formulation."
- This is supported by convincing evidence. The paper provides formal theory for optimization and calculate the Pareto set on this while admitting that there are special cases (the exponential family and multivariate normal distribution). After carefully checked the math, I confirm that it's mathematically sound.

**Claim #3**: "MOVAE, which employs a non-linear decoder to map MVN distributions to real-world distributions, and MOGAN, which learns preference-conditioned generative models."
- This is supported by convincing evidence. The aforementioned theory corroborates the (theoretical) efficacy of the proposed algorithms.

**Essential References Not Discussed:**

N/A

**Experimental Designs Or Analyses:**

See Methods And Evaluation Criteria. My main complaint with the evaluation is that the data they used are quite simplistic. It's hard to tell if such a complex algorithm will scale to a more complex dataset.

**Methods And Evaluation Criteria:**

Based on my understanding, the evaluation metrics are the visualization of the interpolated data.
Strengths:
+ The visualized generated distributions fit nicely the theoretical outcome/results.
Weaknesses:
- The data this paper is evaluated on is quite simple - 2-D tiny greyscale images that are fairly simple and lacking in details (i.e. the tiny images of a microwave just consist of few boxes). It's unclear how this would work or scale to harder images such as CIFAR-10/100.
- The paper motivates the theoretical and algorithmic contributions by claiming that disturbing matching can be used for fields such as domain adaptation, yet the paper didn't use any domain adaption dataset (DomainNet, for example), for interpolation.

**Other Comments Or Suggestions:**

N/A

**Other Strengths And Weaknesses:**

N/A

**Questions For Authors:**

- Could you provide some evidence on how the proposed approach fares against harder data? For example, Office-31 or DomainNet.

**Relation To Broader Scientific Literature:**

This paper could potentially have a very significant impact on broader scientific literature because distribution match is very important in machine learning. Areas such as domain adaptation, generalization, etc. could benefit greatly from it. However, like I have noted in previous comments, there's no evidence that this proposed algorithm will scale to a larger, more difficult website. Esepcially, since the authors are using a variation of the GANs, it might suffer from the issues that the GANs usually face, such as mode collapse.

**Theoretical Claims:**

+ The paper's strongest part is its theoretical contribution. Unless I missed anything, the proof looks right.

---

### Official Review · Reviewer_6Grc · 2025-03-13

**Overall Recommendation:** 3

**Summary:**

The paper propose multiobjective distribution matching (MODM) using tools from information geometry. two concrete algorithms are introduced: a multiobjective variational autoencoder (MOVAE) and a multiobjective generative adversarial network (MOGAN). Experiments on the QuickDraw dataset are provided to demonstrate that the proposed methods are capable of generating high-quality interpolated image distributions.

**Claims And Evidence:**

The claims are supported by mathematical derivations and by experimental results.

**Essential References Not Discussed:**

No.

**Experimental Designs Or Analyses:**

I reviewed the experimental designs and analyses for all experiments in Section 6. There are some minor issues, which will be discussed in Weaknesses.

**Methods And Evaluation Criteria:**

The proposed method are suitable for multi-obejctive distribution macthing.

**Other Comments Or Suggestions:**

For all experiments in the paper, more experimental details should be included, such as the number of epochs, learning rate, and other hyperparameters.

**Other Strengths And Weaknesses:**

**Strengths:**

1. The consideration of multi-objective distribution matching is interesting and novel.
2. The proposed method utilizes rigorous mathematical approaches to solve the problem.
3. The experiments demonstrate the effectiveness of the proposed method.

**Weaknesses:**

1. The proposed method is not compared with the original VAE and GAN on real-world datasets.
2. It appears that no experiments have been conducted using the proposed PrefGAN.
3. The discussion on Multiobjective VAE vs. VAE over Mixture Distributions could be more solid if supported by additional literature and experiments.
4. There are some minor errors in the paper, e.g.,
   - Page 1: "MOVAE (multiobjective Variational Autoencoder)" should be "Multiobjective Variational Autoencoder (MOVAE)."
   - Page 4, line 215: "but letting" should perhaps be "by letting."

**Questions For Authors:**

1. What does Pref-conditioned GAN mean in Figure 3?
2. Why are different preferences used in Figures 3 and 4?
3. How does the runtime of the proposed method compare to the baseline methods?

**Relation To Broader Scientific Literature:**

The keep contribution of the paper realted to multi-objective optimization, information geometry, and generative modeling, which are dissused in the paper.

**Theoretical Claims:**

I reviewed the proofs provided for theorems such as Theorem 8 and Theorem 10 but didn't check the details.

---

### Official Review · Reviewer_nugw · 2025-03-14

**Overall Recommendation:** 2

**Summary:**

This paper studies matching a distribution to multiple target distributions. The authors use information geometry to find Pareto optimal solutions, particularly for the exponential family.  They apply this to multivariate normal distributions and a MOVAE.  They also propose a multi objective GAN.  Experiments demonstrate the performance of MOVAE and MOGAN.

**Claims And Evidence:**

Yes

**Essential References Not Discussed:**

No.

**Experimental Designs Or Analyses:**

Yes.

**Methods And Evaluation Criteria:**

The evaluation criteria is limited. Please see Weaknesses for more details.

**Other Comments Or Suggestions:**

No other comments.

**Other Strengths And Weaknesses:**

Strengths:
1. Considers an important problem that has not been explored enough.
2. Provides novel theoretical framework based on information geometry.

Weaknesses and Questions:
1. One of my major concerns is regarding the limited experimental verification. Only one (arguably toy) image dataset is considered. It is not clear if the method performs well for more complex real-world datasets, such as natural images.
2. The considered experiments do not show the usefulness of the proposed method. Some real use case needs to be included to show the significance of the considered problem and efficacy of the proposed solution.
3. The experimental section lacks quantitative evaluations and baselines. For example, a possible baseline could be using aggregate function to convert multiobjective into single objective optimization.
4. The theoretical analysis is restricted to special families of distributions (mainly exponential families like multivariate normals), which limits the generality. Some discussions on this is needed.

**Questions For Authors:**

Please refer to Weaknesses and Questions

**Relation To Broader Scientific Literature:**

The paper extends core ideas from information geometry and multiobjective optimization (e.g., Pareto set learning and MGDA) to generative modeling and distribution matching.

**Theoretical Claims:**

The theoretical claims appear sound.

---

> ### Author Rebuttal · Authors · 2025-03-30
>
> We thanks for your valuable comments and hope that our following response can address some of your concerns.
>
> **W4 The theoretical analysis is restricted to special families of distributions (mainly exponential families like multivariate normals), which limits the generality. Some discussions on this is needed.**
>
> Actually, exponential family already covers many statistical models, e.g., normal, exponential, log-normal, gamma, chi-squared, beta, Dirichlet, Bernoulli, Poisson, geometric $\ldots$ (see Wikipedia artical “Exponential family”). Meanwhile, our discussion works on dually flat manifolds, which cover most commonly used statistical models including not only exponential families, but also mixture families (convex combinations of distributions), and more generally, $\alpha$-affine manifolds and $\alpha$-families (see Section 3.6 in "Methods of Information Geometry'' by S. Amari and H. Nagaoka). We will complement more concrete examples of exponential families and mixture families in the revised version.

---

> > ### Comment · Reviewer_nugw · 2025-04-09
> >
> > Thanks for the response. Some of my the concerns I raised have not been addressed. Therefore, I keep the score unchanged. I encourage the authors to incorporate the reviewers' comments to increase the quality of the manuscript in future versions.

---

### Official Review · Reviewer_b6z2 · 2025-03-23

**Overall Recommendation:** 3

**Summary:**

The paper develops a theoretical framework for multiobjective distribution matching using information geometry, deriving explicit forms for the Pareto set and Pareto front for exponential family distributions. It applies these insights to design multiobjective generative models.

**Claims And Evidence:**

The derivation of the Pareto set is well supported by theory, but the paper lacks comparisons or ablation studies with conventional methods. Overall, the experimental section is limited and does not provide enough evidence that the proposed approach is more effective than simpler alternatives. For example, a baseline comparison using a vanilla VAE optimized with MSGD would provide a useful reference for readers.

**Essential References Not Discussed:**

N/A

**Experimental Designs Or Analyses:**

There is a lack of comprehensive ablation studies, error bars, or detailed comparisons with alternative multiobjective optimization approaches.

**Methods And Evaluation Criteria:**

The paper primarily demonstrates its concepts through theoretical derivations and illustrative experiments (e.g., with multivariate normal distributions and image interpolation). While this approach makes sense for exploring tradeoffs and Pareto optimality, it can be challenging to directly compare these results without quantitative analysis.

**Other Comments Or Suggestions:**

N/A

**Other Strengths And Weaknesses:**

Strength
- Paper is well written and the theoretical claims and derivations are extensive.
- Extends Pareto optimality in the setting of generative models.

Weakness
- Limited experimental validation, specifically a lack of a good baseline model. Hard to judge practical benefit.
- Rely on strong assumptions that may not work in practice.

**Questions For Authors:**

- Would relaxing the dually flat manifold assumption significantly affect the applicability of the method in practice?
- How scalable are both MOVAE and MOGAN with respect to the dimension of the dataset?

**Relation To Broader Scientific Literature:**

This theoretical contribution builds on and complements prior work in multiobjective optimization and generative modeling.

**Theoretical Claims:**

To the best of my knowledge the theoretical claims and proofs in the appendix seems to be correct and extensively derived.

---

> ### Author Rebuttal · Authors · 2025-03-30
>
> We sincerely thanks for all valuable comments from you. We thanks for that your believe our theoretical results is strong and hope that our following response can address your concerns more or less.
>
> ----
>
> **W1. Limited experimental validation.**
>
> To response with that, we have included a new dataset called ageing of real world images.
>
> **W2. Rely on strong assumption that may not work in practice**
> We address this issue in two parts. First, we derive the explicit Pareto set formulation for the dually-flat manifold, encompassing the Exponential and mixture families. This applies to a wide range of distributions, including MVN, Poisson, Gamma, Wishart, Beta, and Hypergeometric. Second, we extend our approach using nonlinear models like VAE and GAN to transform MVN into complex real-world distributions, further broadening its applicability.
>
> **Q1. Would relaxing the dually flat manifold assumption significantly affect the applicability of the method in practice?**
>
> In fact, most of the commonly used statistical models can be covered by the case of dually flat manifold, e.g., MVN model, Possion model, Gamma model, probability simplex ... If more generally, a non dually flat statistical models is considered, similar analysis can still be applied to its canonical divergence function, but there will be an additional error term of 4th order in the result (see Section 3.8 in "Methods of Information Geometry'' by S. Amari and H. Nagaoka).
>
> **Q2. How scalable are both MOVAE and MOGAN with respect to the dimension of the dataset?**
>
> We use MOOVAE as an illustrative example. In MOOVAE, the decoder network generates the output image from a latent vector $ z $, which typically has a low-dimensional representation. The Pareto-optimal distribution is constructed within this latent space and subsequently decoded into an image. By keeping the latent vector relatively small, the model maintains efficiency. The scalability for generating larger images primarily depends on the capacity and power of the neural network.
>
> To empirically show the scalability power, we also have added a new experiments on the ageing dataset, where the images are realworld $3 \times 512 \times 512$ images.
>
> ---
> Reference
>
> [1] https://github.com/royorel/FFHQ-Aging-Dataset.

---

### Decision · Program_Chairs · 2025-05-01

**Decision:**

Accept (poster)

**Comment:**

This paper focuses on the problem of "generating a distribution that aligns with multiple underlying distributions". The core contribution of the paper. The core contributions are characterizing the pareto optimal set in case of exponential family. The paper also has empirical section that shows the practical implementation of the proposed schemes. The reviewers appreciated theoretical contribution but find empirical comparison a bit underwhelming. Overall, this is a good paper to be accepted.